# A rationally engineered decoder of transient intracellular signals

Claude Lormeau [1,2], Fabian Rudolf[1] & Jörg Stelling [1✉]

Cells can encode information about their environment by modulating signaling dynamics and responding accordingly. Yet, the mechanisms cells use to decode these dynamics remain unknown when cells respond exclusively to transient signals. Here, we approach design principles underlying such decoding by rationally engineering a synthetic short-pulse decoder in budding yeast. A computational method for rapid prototyping, TopoDesign, allowed us to explore 4122 possible circuit architectures, design targeted experiments, and then rationally select a single circuit for implementation. This circuit demonstrates short-pulse decoding through incoherent feedforward and positive feedback. We predict incoherent feedforward to be essential for decoding transient signals, thereby complementing proposed design principles of temporal filtering, the ability to respond to sustained signals, but not to transient signals. More generally, we anticipate TopoDesign to help designing other synthetic circuits with non-intuitive dynamics, simply by assembling available biological components.

[1] Department of Biosystems Science and Engineering and SIB Swiss Institute of Bioinformatics, ETH Zurich, Mattenstrasse 26, CH 4058 Basel, Switzerland. [2] Life Science Zurich Graduate School, Interdisciplinary PhD Program Systems Biology, Zurich, Switzerland. ✉email: joerg.stelling@bsse.ethz.ch

Cells can shape the dynamic responses of signaling pathways to encode information about their environment, which then requires an interpretation of the resulting dynamics (decoding) to elicit appropriate responses, for example, in terms of gene expression programs[1]. The mammalian MAPK pathway is a prominent example for such dynamic encoding and decoding in cellular signaling. It responds to NGF with a sustained Erk output to induce differentiation, but to EGF with a transient output to induce proliferation[2]. A coherent feedforward (CFF) on c-Fos, a network motif in which a signal activates the target both directly and via an intermediary component[3], decodes the sustained output of MAPK signaling[4,5]. This architecture is consistent with proposed design principles of temporal filtering, which is the ability to respond to sustained signals, but not to transient signals[4,6].

However, it is unknown how cells decode the transient output of MAPK signaling[7]. Corresponding mechanisms are unlikely to reside only in the dynamics of a single promoter[8]; they are rather established by interaction networks that are not yet identified. More generally, beyond suggested signal processing with cooperative assemblies[9], decoding mechanisms to generate a specific response to a transient signal, while ignoring more sustained signals and not responding without input, are currently unknown.

In addition to analyzing the natural system, the design of simple synthetic circuits with the same phenotype can help decipher complex phenotypes and extract the underlying principles from which they emerge[10]. A comparatively simple signaling pathway to investigate decoding principles by synthetic circuit design is the mating pathway in the budding yeast *Saccharomyces cerevisiae*. It is well-characterized, accepts a sustained stimulation with the α-factor pheromone as input[11], and was previously used, for example, to rationally tune G-protein coupled receptor signaling in this model eukaryote[12].

In this work, we elucidate decoding mechanisms by rationally designing a synthetic short pulse decoder for the budding yeast mating pathway (Fig. 1a). Specifically, we engineer a circuit that responds to a 30 min pulse of α-factor, but not to no pulse or a 3 h pulse, which is orthogonal to the natural mating response. Because decoder network architectures are not known, we develop a computational method for rapid prototyping of synthetic circuits with complex target dynamics, TopoDesign. We show that the method can explore thousands of possible circuit architectures, design targeted experiments, and then rationally select a single circuit for implementation. Our implemented circuit demonstrates short pulse decoding through incoherent feedforward (IFF) and positive feedback (PF), and we predict nested IFF loops to be essential for decoding transient signals more generally.

## Results

**Topological design framework.** Because we did not know which network structures (topologies) could generate a short pulse decoder behavior, we defined a master network (Fig. 1b) that encompasses well-known motifs in signal processing: negative and positive feedback (NF and PF)[13,14] as well as IFF[3,15] motifs. We included IFFs in particular because they can retain memories of pulses[16], and thereby discriminate between transient and sustained inputs[17]. Note, however, that we require a decoder behavior that is different from the known behaviors of IFFs: it should not respond to a long pulse at any point in time, not only after an adaptation period.

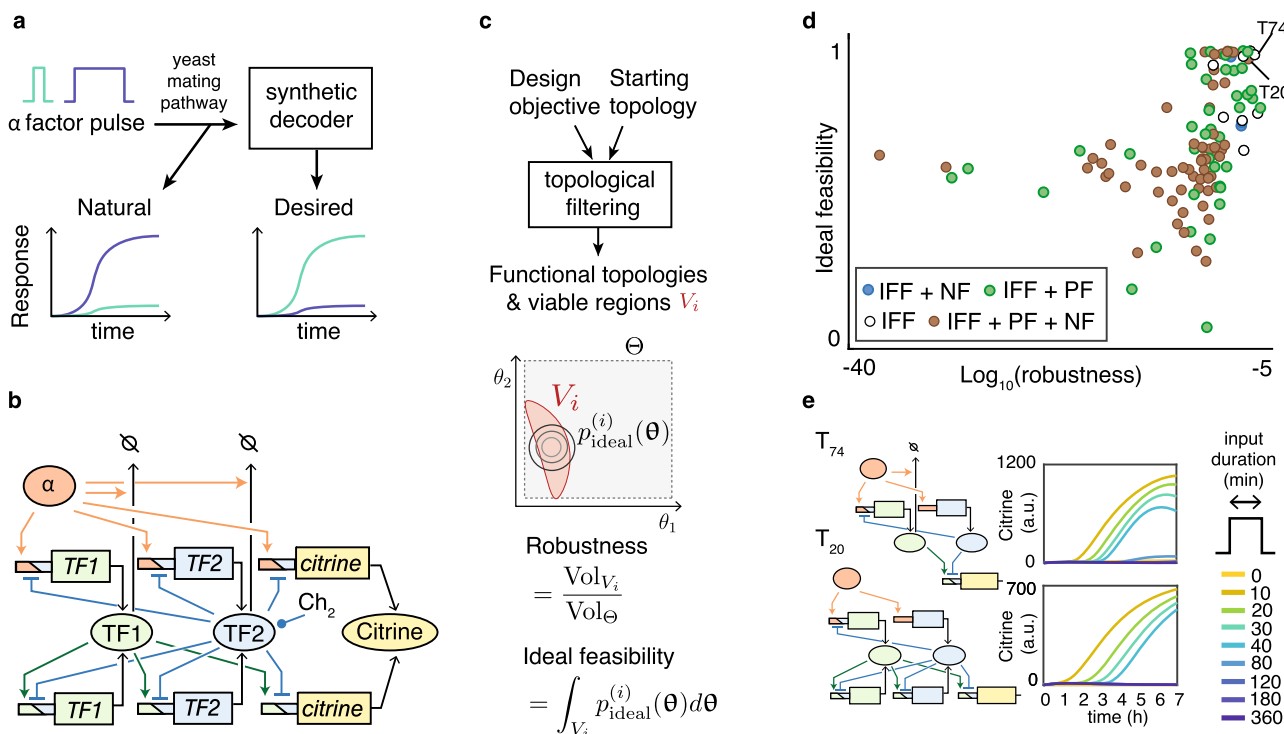

**Fig. 1 Initial decoder design. a** Design objective for a short pulse (≤30 min duration) decoder of signals transmitted by the yeast mating pathway. **b** Starting topology with α-factor and chemical (Ch₂) input, citrine fluorescence output, transcription factors (TF1,2), combinatorial promoters (trapezoids), and activating (colored arrows) or inhibiting (bars) interactions; see also Supplementary Methods. **c** First step of TopoDesign, requiring a design objective (**a**) and a starting topology (**b**) for topological filtering[25] to obtain functional topologies $T_i$ by deleting interactions from the starting topology; they achieve the design objective for at least one parameter set **θ** (individual parameters: $\theta_i$). Viable regions $V_i$ in parameter space (boundaries **Θ**) allow to compute robustness and feasibility metrics inspired by[23], where $p_{ideal}^{(i)}(\theta)$ is an ideal parameter distribution. **d** Ranking of the 109 topologies that can meet the design objectives, comprising incoherent feedforward (IFF), negative feedback (NF), and positive feedback (PF) motifs. **e** Topologies highlighted in **d** and their predicted behaviors (random viable parameter sets).

In biological terms, the master network (Fig. 1b) involves an activating transcription factor (TF1), a repressor (TF2) that responds to a tuning chemical, six potential AND gate inducible promoters (activated only if TF1 is present and TF2 is absent), and two post-translational interactions inspired by ref. [18]. At this stage, the biological parts are hypothetical—they exist in principle, but we do not know if specific instances with suitable quantitative characteristics, such as repressor strength are available or will need to be established by molecular engineering.

Enumeration of the master network's sub-topologies yields a set of 4122 possible decoder topologies. Experimentally testing that many alternatives is impossible, although efficient methods for circuit construction exist[6,9,13,14,19,20]. Also, existing computational design methods either help designing dynamic circuits for few topologies that can be enumerated and analyzed individually[21], or focus on logical circuits using well-characterized components[20,22]. Because neither condition applied here, we extended our Bayesian circuit design method[23] to a rapid prototyping method, TopoDesign. Without requiring a catalog of specific biological parts, it explores possible combinations of hypothetical parts. TopoDesign accounts for uncertain knowledge about the parts and their biological variability, which allows capitalizing on existing biological parts that may be ill-characterized.

We define a design objective and a dynamic mathematical model for the set of possible topologies (Fig. 1c, Supplementary Fig. 1; Supplementary Methods). For the model, we used commonly applied specifications of processes and interactions, such as Hill functions for gene expression control; note that all model inferences and predictions are contingent on this formulation. Topological filtering[24,25] explores simultaneously model topologies and parameters to find functional topologies $T_i$. For each $T_i$, at least one parameter set is viable: it achieves the design objective. By efficient sampling[26], we also obtain the corresponding viable space $V_i$, that is, the parameter space for correct circuit behavior. This allows us to define two metrics for topology robustness and feasibility (Fig. 1c). Our robustness metric is a quantitative version of the "Q-value" or "robustness score"[4,27]. It measures globally how much of the parameter space is viable, giving the theoretical probability of a circuit to achieve the design objective, without prior knowledge on parameters. However, with correlated parameters, a topology may not tolerate variation in individual parameters. To account for such dependencies between parts in practice, we therefore measure feasibility: the proportion of a parameter distribution that fits into the viable space (Fig. 1c). The definition of the metric is distinct from[23] and critical: it enables a systematic integration of experimental data, and thereby all iterations of computation and experiments. Without further information, we compute ideal feasibility by assuming optimal parameters with small variance (Fig. 1c; Supplementary Methods).

**Functional topologies for a short pulse decoder**. The search for functional topologies is unbiased by experimental data, assuming only broad, plausible ranges of parameters (Supplementary Methods). It yields both a set of functional topologies that one can analyze to reveal principles of decoder function and (via the viable spaces) constraints on the characteristics of parts to be used for circuit implementations. Specifically, TopoDesign found 109 topologies able to behave as short pulse decoders (Fig. 1d, Supplementary Figs. 2 and 3). Robustness and ideal feasibility correlate only moderately (Kendall's $\tau = 0.40$, $p < 10^{-9}$), supporting the need for two metrics to capture fully the size and shape of the viable spaces. All circuits include at least one IFF motif. Some have an additional NF, PF, or both. Apart from the omnipresent IFF, the diversity of motifs in robust and feasible topologies does not orient us towards one particular circuit architecture. For

example, $T_{20}$ and $T_{74}$ are indistinguishable in our metrics (Fig. 1d), but rely on very different topologies to generate similar predicted decoder behaviors (Fig. 1e).

To understand how functional circuits decode the input dynamics, we simulated the internal dynamics of four simple circuits with a random viable parameter sample each (Fig. 2). At the core of each circuit are interlocked incoherent (comprising α factor, TF1, and TF2) and coherent (TF2, TF1, and citrine) feedforwards (for a more detailed analysis of the relations between network motifs and decoder function, see also Supplementary Fig. 3 and Supplementary Methods). In $T_{30}$ as the minimal example, the IFF on the activator TF1 generates an adaptive pulse of TF1 activity that has approximately the duration of the input pulse we want to decode, independent of the input duration. This adaptive TF1 response is crucial since all 109 circuits except one (with very low robustness) include an IFF on TF1. Because the negative regulator TF2 always follows the input with a pulse that has approximately the same duration as the input, citrine appears only if TF2 disappears before the TF1 signal disappears, hence only in response to short inputs. Circuits with additional interactions employ the same principle (Fig. 2), with additional increases of TF1 activity, for example, due to PF in $T_{39}$ that stabilizes the output in a high steady state.

**Specification of biological parts and rapid prototyping**. To specify biological parts, in principle, one can select parts from established catalogs that are expanding in scope also for *S. cerevisiae*[22,28]. The bottleneck is that quantitative parts characterizations that allow checking if parts fulfill the requirements on parameters for circuit function (via the viable spaces) remain sparse. TopoDesign, however, can also consider less well-characterized components by explicitly accounting for parameter uncertainties. We decided to use our available biological components; we matched them to model predictions by identifying recurrent constraints on parameters in circuits with ideal feasibility >0.9 (Fig. 1d, Supplementary Fig. 4). They required TFs with relatively high fold change and cooperativity ($n > 2$). We selected a corresponding activating TF1 (Fig. 1b): LexA-ER-B112 acting on a target promoter with four lexA boxes[29]. Similarly, we specified TF2 by a TetR-MBP fusion protein repressing a Tdh3 promoter flanked by tetO sites[30], tunable by anhydrotetracycline (aTc), and we used the native, α-factor-inducible Fus1 promoter. Inducible promoters could be hybrid: repressed by TetR-MBP and activated by α-factor ($P_{\text{fus1tet}}$) or LexA-ER-B112 ($P_{\text{lexAtet}}$). To establish α-factor-responsive protein degradation, we tagged TF1 with a phospho-regulon[18] (few functional topologies included controlled TF2 degradation).

Different configurations of these few parts could yield 69 functional topologies. To select among them rapidly to reduce experimental effort, we used prototyping, namely construction of small informative synthetic networks. We built seven such networks and measured their dynamics and dose-responses to aTc and α-factor (Fig. 3a–c, Supplementary Fig. 5). With 984 data points from flow cytometry in total and a uniform prior (Supplementary Table 7), we used approximate Bayesian computation[31] (ABC) to estimate the posterior probability distribution of the 20 parameters for all parts, and thereby to make the parts' characteristics usable for the evaluation of the 69 candidate topologies (Fig. 3d; Supplementary Methods). Narrow distributions (Fig. 3e, Supplementary Fig. 6) indicate that the data suffice to obtain high-quality information on all parameters.

We used the ABC posterior to update the feasibility of all circuits (Fig. 4a; Supplementary Fig. 2). Because the nonzero regions of the posteriors did not overlap with the topologies' viable spaces, all circuits had zero updated feasibility. To increase

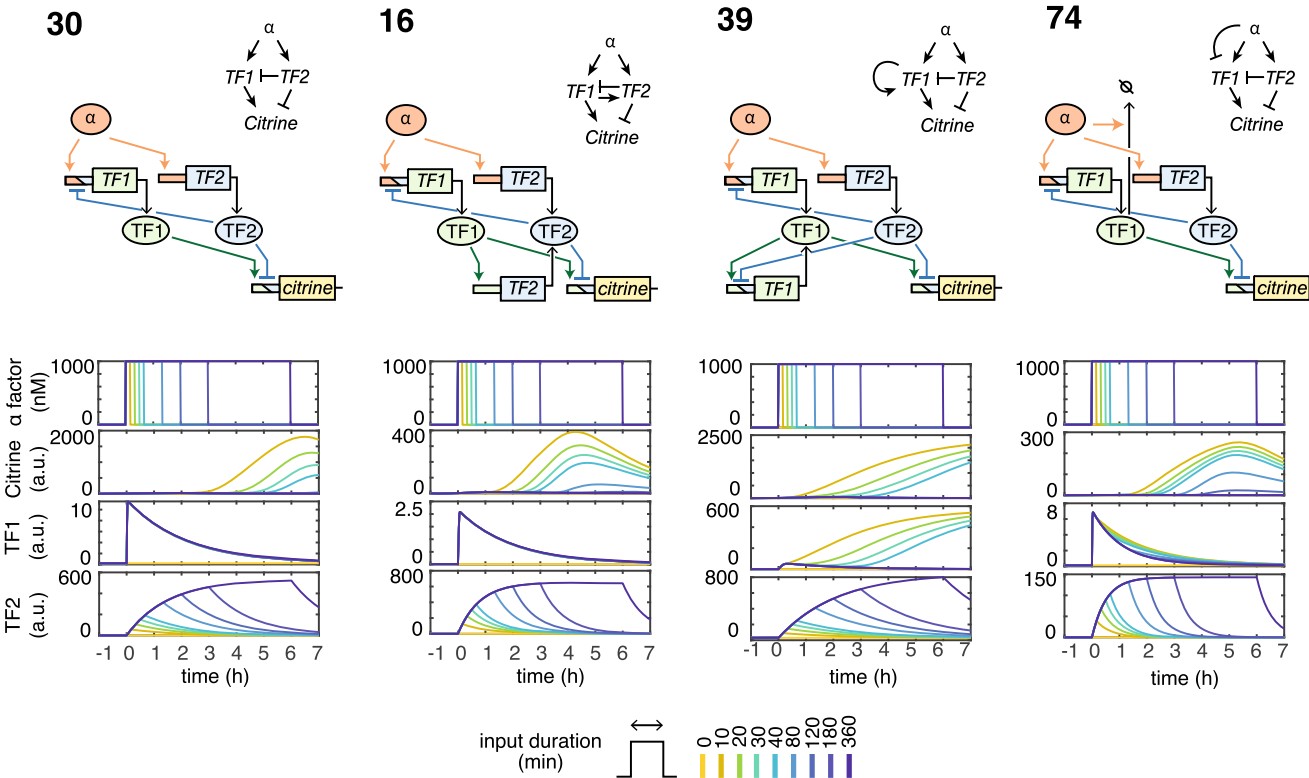

**Fig. 2 Decoder design principles.** Detailed (colors; see Fig. 1) and abstracted (black) topologies of four simple functional circuits (top; numbers indicate topology variants). Internal dynamics of TF1 and TF2 were simulated for a random viable parameter sample for each circuit, with different input (α-factor) durations (indicated by colors; bottom).

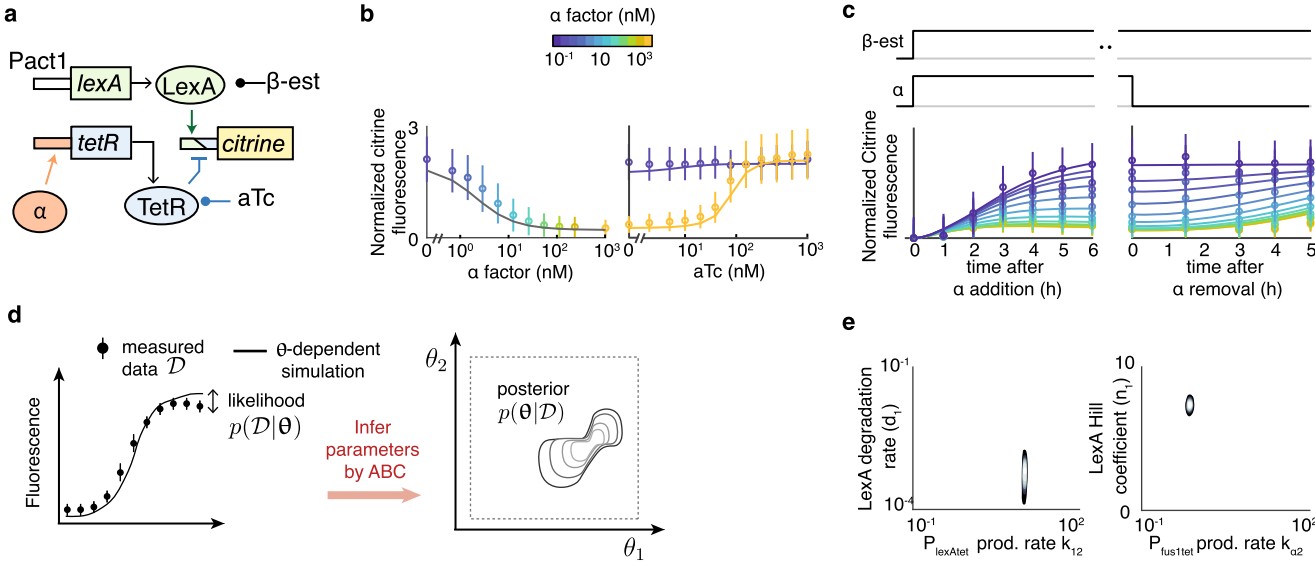

**Fig. 3 Rapid prototyping. a** Example module for characterization (strain yCL109; β-est: β-estradiol; aTc: anhydrotetracyclin). **b, c** Characterization experiments, citrine fluorescence measured by flow cytometry. All flow cytometry measurements include at least 4000 cells after gating. Symbols show experimental means, ±standard deviation, and lines simulations of the maximum likelihood parameter set estimated with all seven modules. 5 μM β-estradiol were added at time 0, together with varying concentrations of α-factor (dose response at 18 h (**b**), dynamics after α addition (**c**)), or varying concentrations of aTc with 0 or 1 μM α (aTc dose response at 6 h (**b**)). For α-factor release (**c**), we removed α from the medium 18 h after induction. **d** Illustration of the second step of TopoDesign to infer the posterior distribution $p(\theta|D)$ of the 20 parameters $\theta_i$ for parts in Fig. 1b by approximate Bayesian computation (ABC) using the likelihood $p(D|\theta)$ for data $D$. **e** Projection of the joint posterior parameter distribution on two pairs of parameters; bright contour lines indicate high probability density.

feasibility, we allowed for the tuning of selected parameters. Tunability is an intrinsic property of a parameter, reflecting the experimental effort for modifying a parameter's value in given ranges. We devised discrete categories of tunability that account for this effort and have different roles in TopoDesign (Supplementary Methods). As simple tuning possibilities to increase feasibility, we considered varying aTc concentrations and promoter copy numbers (assuming those affect maximum production rates proportionally).

After optimizing the location of the parameter posterior in the tuning directions, feasibility clearly discriminated between the 69 topologies (Fig. 4b). Two circuits, $T_{39}$ and $T_{93}$, stood out by having close to 50% feasibility. $T_{39}$ (Fig. 4c) was more robust and had fewer interactions than $T_{93}$ (Supplementary Fig. 2). Importantly, the additional induced degradation of LexA in $T_{93}$ is likely to increase the circuit's burden on the cell and thereby to make it evolutionarily less stable[32]. Finally, tuning without molecular engineering to modify parameters, such as the cooperativity of LexA should make $T_{39}$ a functional decoder (Fig. 4d, Supplementary Figs. 6 and 7). We therefore selected $T_{39}$ for implementation.

**Circuit implementation and validation.** To decide how to construct $T_{39}$ (Fig. 5a), we explored the parameter subspace for copy number variations of the individual parts and aTc concentration; they are experimentally simple to control and therefore suitable for rapid prototyping. The sampling results (Supplementary Fig. 8) indicated that the two $P_{lexAtet}$ constructs required single copies; we integrated single copies using shuttle vectors from ref. [33] to obtain strain yCL114 (Supplementary Table 2). The model also predicted a need for high copy numbers of the two α-factor-inducible constructs, and enough aTc to substantially reduce TetR binding for a working decoder. For those constructs, we implemented variants of $T_{39}$ with variable copy numbers, determined a posteriori. Specifically, we cloned each part in a multi-integration vector deficient in auxotrophic marker production and transformed a mixture of the two resulting constructs in strain yCL114 (see "Methods" section).

Next, we used our α-factor-pulse flow cytometry assay at 100 nM aTc (see "Methods" section) to screen for cells that did not respond without α-factor or to a long pulse, but responded to a short pulse. This yielded four strains with circuit variants $T_{39.1-4}$ (see Fig. 5b for $T_{39.2}$) that could behave as short pulse decoders. Indeed, in independent experiments in which we systematically varied the pulse duration, $T_{39.1-4}$'s responses increased and then decreased again with pulse duration (Fig. 5c). In contrast, circuit variants with only a PF and no IFF, which we constructed as negative controls ($C_{1-3}$, Fig. 5a), were non-functional (Fig. 5c), as expected from their lack of IFFs.

To characterize circuit behaviors with respect to the aTc concentration as tuning parameter, we varied aTc concentrations and compared the responses of all strains to 30 min and 3 h pulses

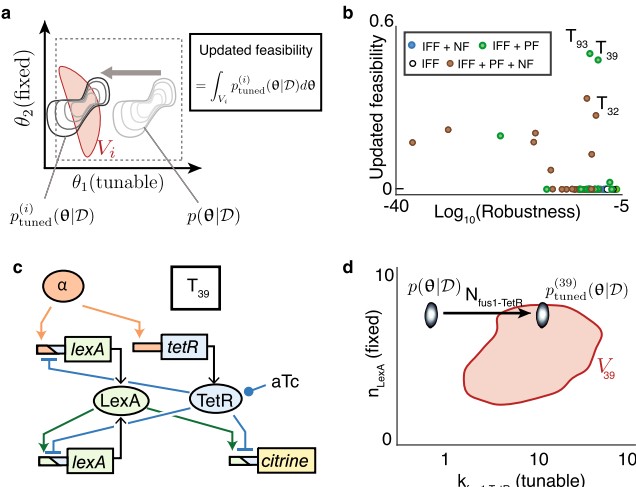

**Fig. 4 Bayesian updating. a** The inferred parameter posterior $p(\theta|D)$ is compared to the viable regions $V_i$ of all topologies to calculate each topology's feasibility; we shift $p(\theta|D)$ in tunable parameter directions to $p_{tuned}^{i}(\theta|D)$ to maximize feasibility. **b** Updated ranking of topologies for the short pulse decoder, highlighting promising topologies. **c** Circuit diagram of the best candidate, $T_{39}$. **d** Projection of the viable space of $T_{39}$, and of the parameter posterior distribution before and after tuning, on two parameters. Copy number $N_{fus1\,TetR}$ enables tuning of promoter strength, leading to a predicted feasibility of 48% (**b**).

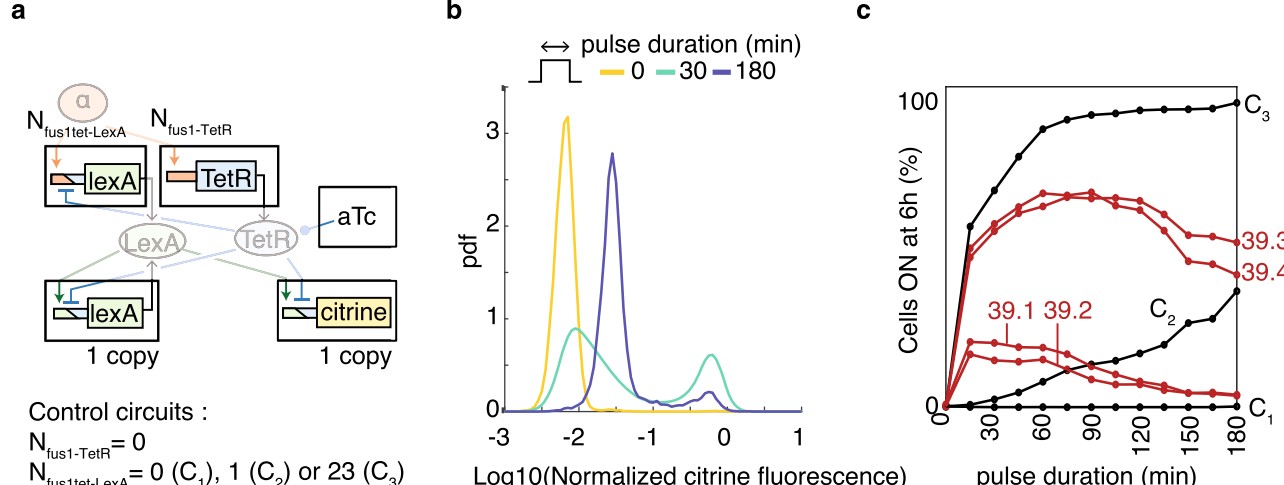

**Fig. 5 Implementation of the predicted decoder. a** Variants of $T_{39}$ differ in the copy number $N$ of α-factor inducible TetR and LexA constructs. Control circuit variants ($C_{1-3}$) lack α-factor inducible TetR. **b** Distributions of single-cell responses of $T_{39.2}$ to 1 µM α-factor pulses (durations indicated by colors) at 100 nM aTc as determined experimentally by flow cytometry. All flow cytometry measurements include at least 4000 cells after gating. Fluorescence is normalized to the FSC-A signal, see "Methods" section. **c** Responses of the indicated $T_{39}$ variants and control circuits at 6 h to different 1 µM α-factor pulse durations at 100 nM aTc in percentages of cells above a fluorescence threshold (unimodal response of $C_3$ to a 3 h pulse is 95%), to accommodate for bimodal distributions.

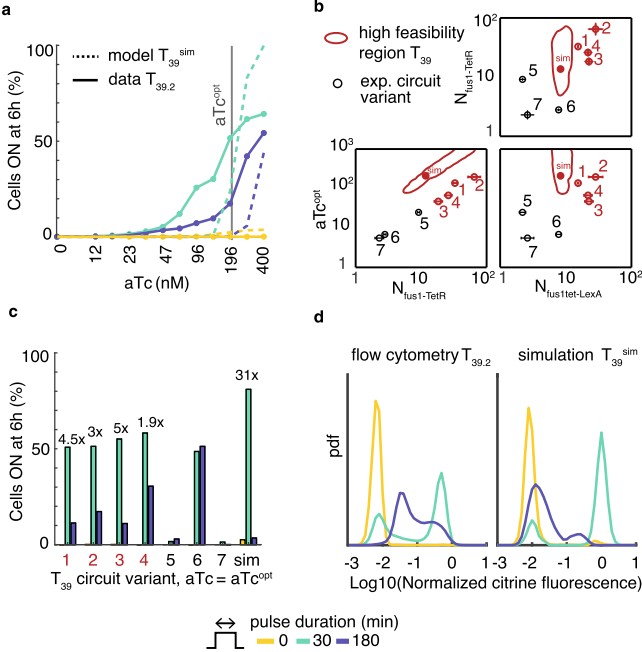

between 30 min and 3 h pulses between twofold and fivefold (Fig. 6c). To estimate the predicted behavior in a cell population, we performed simulations of $T_{39}$ ($T_{39}^{sim}$) with copy numbers and aTc concentration optimized for feasibility (Fig. 6b), and each simulated cell otherwise parametrized with samples from the ABC posterior distribution (see Supplementary Methods for details). Given the definition of feasibility, half of these simulated cells achieve the design objective individually. In this best-case scenario, $T_{39}^{sim}$ showed a similar qualitative behavior as the experimental data (Fig. 6a) and the predicted fold change was 31 (Fig. 6c). This higher value compared to the experimental data resulted primarily from fewer cells responding to the 3 h pulse. A comparison of flow cytometry data and simulation results (Fig. 6d) pointed to an explanation: measured and simulated distributions for $T_{39}$ had similar shapes, but a different prominence of the two modes for the 3 h pulse. We speculate that a combination of gene expression noise (not represented in our models) and PF stabilizing the high output state (indicated by the control circuit's unimodal distribution) causes this difference. Overall, hence, predicted functional circuits not only achieved the non-intuitive qualitative behavior but also met the quantitative design objectives within a meaningful margin of the best-case scenario.

## Discussion

Our decoder demonstrates that a purely transcriptional circuit with four nodes combining an IFF with a nested CFF and a PF can respond exclusively to short inputs. This is an example for how nature could discriminate between input durations, but not the only solution (Supplementary Fig. 3): our topology exploration predicted many other possibilities. Intriguingly, they all featured an IFF, pointing to a common design principle. Specifically, we expect corresponding circuits to have nested IFFs at their core because an adaptive response is required to discriminate between short pulses on the one hand, and no or long pulses on the other hand. Either an IFF itself, or a time-delayed NF embedded in an IFF can generate the critical adaptive response (Supplementary Fig. 3 and Supplementary Methods); NFs and IFFs are the two known network motifs that can achieve adaptation[27]. However, we cannot exclude that more complex decoder architectures exist.

We argue that a perspective in terms of network motifs can help identify and explain naturally occurring decoders of short inputs. For example, interlocked feedforwards occur frequently in gene regulatory networks involved in the development of multicellular eukaryotes, such as *Drosophila*[34]. Intriguingly, short and long pulses of Erk activity lead to the specification of distinct cell types during fly development[35], but the mechanisms for the short pulse response are unclear. Similarly, many feedforwards are known in mammalian Erk signaling[36] and downstream gene regulation can decode Erk dynamics[37]. Candidates for short pulse decoding are the c-Fos transcription factor and the mRNA-destabilizing protein ZFP36 involved in an IFF[38], or Erk and dual-specificity phosphatases involved in time-delayed NF as well as IFF[36], provided that these regulators act antagonistically on common targets.

More generally, TopoDesign combines Bayesian accounting for uncertainty in design[21] with scalability in terms of the number of possible topologies, relevant metrics for selecting topologies, and rapid prototyping to reduce experimental effort. Scalability with respect to circuit complexity (dimensions of parameter spaces), however, is an open issue for future investigations. In addition, one could develop model-based experimental design approaches to identify small informative networks during rapid prototyping, and expand the framework to account for cell-to-cell variability explicitly. While being general, TopoDesign is customizable: the

**Fig. 6 Decoder validation and performance. a** Responses at 6 h to 0, 30, or 180 min pulses (colors) of 1 μM α-factor for $T_{39.2}$ at varying aTc concentrations as determined experimentally by flow cytometry (see Supplementary Fig. 9a for all circuits). All flow cytometry measurements include at least 4000 cells after gating. Experimental data (solid lines) are complemented by model predictions (dashed lines) for an optimal $T_{39}$ implementation ($T_{39}^{sim}$, see Supplementary Methods). **b** Experimentally determined copy numbers for seven implemented variants of $T_{39}$ (mean ± s.d., $n = 3$ technical replicates) and optimal aTc concentrations (aTc$^{opt}$; estimated for $T_{39.1-4}$ (red open symbols) and extrapolated for $T_{39.5-7}$ (black open symbols; see also Supplementary Fig. 9b)). Red contour line: predicted region of high feasibility. Filled red symbol: copy numbers and aTc concentration for $T_{39}^{sim}$. **c** Responses at 6 h for $T_{39.1-7}$ and for the optimal simulated circuit at aTc$^{opt}$. Numbers above bars: fold-changes in responses to 30 min pulses relative to 180 min pulses. **d** Distributions of single-cell responses of $T_{39}$ (pdf: probability density function) to input pulses at aTc$^{opt}$, determined experimentally as in Fig. 5b (left) and predicted computationally (right; equivalent normalization).

of α-factor. This experiment confirmed that $T_{39.1-4}$ could operate as short pulse decoders, whereas $C_{1-3}$ could not (Fig. 6a for $T_{39.2}$, Supplementary Fig. 9a for all variants). It also identified variant-specific optimal aTc concentrations (see Fig. 6a, aTc$^{opt}$: highest 30/180 min response ratio, at least 50% of cells responding).

To compare model predictions and experimental data for the $T_{39}$ circuit design, we then used the sampling results (Supplementary Fig. 8) to define a region of high feasibility (>90% of $T_{39}$'s maximal feasibility) in the space of tunable parameters (Fig. 6b). $T_{39.1-4}$ differ in copy numbers, which we determined by qPCR. The combined data suggest that $T_{39.1-4}$ form a region with the same shape as the predicted feasibility region, although it is shifted (Fig. 6b). This is supported by aTc$^{opt}$ of $T_{39.1-4}$ correlating with the number of TetR constructs (Pearson's $r = 0.92$, $p < 10^{-4}$, Supplementary Fig. 9b) as expected. To test the hypothesis of a functional region, we selected three additional strains ($T_{39.5-7}$) with high copy numbers according to qPCR results, but located outside the region of $T_{39.1-4}$ (Fig. 6b). We predicted $T_{39.5-7}$ to be non-functional, which experiments with varying aTc concentration confirmed (Supplementary Fig. 9a).

Finally, we evaluated if the experimental decoder performance is consistent with model predictions. For $T_{39.1-4}$ at their specific aTc$^{opt}$, we measured a fold change of response for discriminating

user defines available components and tunable parameters. Similar to our study, we envisage that TopoDesign will accelerate the engineering of synthetic circuits with complex dynamic behavior, without detailed molecular engineering.

## Methods

**Plasmid construction.** Plasmids (Supplementary Table 1) were constructed by isothermal assembly using the pRG shuttle vector series[33] as backbones, and inserts were obtained by PCR. Primers used for plasmid assembly are listed in Supplementary Table 4. All constructs were checked with Sanger sequencing (Microsynth). The sequences of the three hybrid promoters cloned for this work are listed in Supplementary Table 3. The fus1tet and lexAtet promoters were obtained by fusing the core promoter sequence of P2tet from Azizoglu et al.[30] to the upstream activating sequence of either the Fus1 promoter or the 4 lexA boxes promoter from Ottoz et al.[29]. Instead of using directly the Fus1 promoter when no repression by TetR was needed, a non-repressible version of the Fus1tet fusion promoter (called Pfus1mut) was used to keep exactly the same properties as Pfus1tet. The fus1mut promoter is almost the same as the fus1tet promoter except that the sequences of the tetO sites were shuffled to prevent binding of TetR. The LexA-ER-B112-phosphodegron was obtained by inserting the phosphodegron sequence from Grodley et al.[18] between the end of the LexA-ER-B112 sequence from Ottoz et al.[29] and the stop codon.

**Yeast strain construction.** *S. cerevisiae* strains are listed in Supplementary Table 2. They were constructed for this work except for FRY69[33], from which yCL102 was derived with the modifications bar1::Nat and far1::KanMX. All other strains were then derived from yCL102. Integration of single-copy constructs was always done with one of the pRG20x vectors[33], and checked for single-copy integration at the correct site by multiplex colony PCR (protocol by Gnügge et al.[33]). Integration of multiple copy constructs was done with the pRG235 vector, with a co-transformation if different constructs had to be integrated in multiple copies. The number of integrated copies for each construct was then checked by quantitative real-time PCR (qRT-PCR).

**Media and chemicals.** All experiments were performed at 30° in YPD medium containing 1% yeast extract (Thermofisher, 212720), 2% bacto-peptone (Thermofisher, 211820) and 2% glucose (Sigma, G8270).

α-factor mating pheromone (Zymo Research, Y1001) was directly used as 10 mM stock. aTc (Cayman Chemicals, 10009542) was prepared as a 10 mM stock in ethanol. β-estradiol (Sigma-Aldrich, 107K1322) was prepared as a 100 mM stock in ethanol. Pronase (protease from Streptomyces griseus, Sigma-Aldrich, P8811) was prepared as a 40 mg/ml stock solution in sterile distilled water.

**Flow cytometry.** For all experiments we first cultured cells to early exponential phase (about 5e6 cells/mL), then we added β-estradiol to reach a concentration of 5 μM. For aTc dose-response experiments, α-factor (final concentration 1 μM) and aTc (various concentrations) were added together with β-estradiol. For α-factor dose-response experiments, only α-factor (various concentrations) was added together with β-estradiol. For α-factor release experiments, cells were diluted when adding β-estradiol and α-factor to make sure they are in exponential phase 18 h later. 18 h after adding β-estradiol and α-factor, cells were taken to a Corning FiltrEx 96-well white filter plate with 0.2 μm hydrophilic PVDF membrane to be centrifuged for 3 min at 3000 g. They were then resuspended in new medium with 50 μg/mL pronase (to remove the remaining α-factor) and 5 μM β-estradiol (no α). For α-factor pulse experiments, α-factor was removed from an aliquot of the main culture by centrifugation as for α-factor release experiments after each pulse duration, and aTc was added to each aliquot after α-factor removal in the indicated concentration together with pronase and β-estradiol.

For all experiments, at every time point indicated, cells were diluted in PBS and measured using a BD LSR Fortessa cell analyzer equipped with a high-throughput sampler. PMT voltages used for the different channels were always 480 mV for forward scattering, 275 mV for side scattering, and 630 mV for the 488 nm excitation laser. A 530/30 filter was used to measure Citrine fluorescence. We gated broadly for budding cells in the FSC-W-SSC-W plane[29] as shown in Supplementary Fig. 10. Our cells do not stop growing in the presence of α-factor due to *far1* deletion, but they tend to aggregate instead, leading to a higher fluorescence signal in the presence of α even for a constitutive promoter. In order to correct for the size of flow cytometry events, we normalize the fluorescence of every event by its FSC-A signal. The normalization corrects for the α-factor effect on cell size. However, we still observe a small shift induced by α-factor for the act1 strain (yCL106). We included this unexplained effect in our model of the act1 promoter (see methods about computing the parameter posterior in Supplementary Information).

**Quantitative real-time PCR.** qRT-PCR was used to assess the number of copies of *lexA-ER-B112*, *tetR-nls-malE* and *citrine* constructs in the genome for the strains yCL110, yCL130-133, and yCL141-143. We extracted genomic DNA of dense cultures with a YeaStar Genomic DNA Kit with Zymo-Spin III columns

(Zymo Research). We performed qRT-PCR on a LightCycler 480 Instrument using the PowerUp SYBR Green Master Mix (ThermoFisher) with primers listed in Supplementary Table 5. We fitted all fluorescence curves with the five-parameter logistic curve from[39] and used the analytical solution of the second derivative maximum to determine the Ct values. For copy number quantification, we used citrine (always present as one copy) as an internal reference and we used the strain yCL110 (containing one copy of each target) as a reference strain. Primer efficiencies $E_{LexA}$, $E_{Citrine}$, and $E_{TetR}$ were estimated with calibration curves, and copy numbers were estimated as in Pfaff et al.[40].

**Software for data collection and analysis.** BD FACSDiva 8.1 software was used to collect the flow cytometry data. Roche LightCycler® 96 SW 1.1 software was used to collect the RT-qPCR data. Matlab R2019a (Mathworks, Natick, MA) was used to analyze the data, and to develop the TopoDesign method described in detail in Supplementary Methods. To analyze FACS data, we used the toolbox MatlabCytofUtilities available from https://github.com/nolanlab/MatlabCytofUtilities. The TopoDesign method depends on the Matlab toolboxes Hyperspace (https://gitlab.com/csb.ethz/HYPERSPACE, commit of 09/17/2018), TopoFilter v0.3.6 (https://git.bsse.ch/csb/TopoFilter), IQM Tools v1.2.2.2 (https://iqmtools.intiquan.com/) and the 2014 MEIGO-M package (available from http://gingproc.iim.csic.es/meigom.html).

**Statistics and reproducibility.** Each experiment was repeated independently at least three times with similar results, with the exception of the dynamic response of small informative networks (columns 3 and 4 in Supplementary Fig. 5) which was carried out once, and quantitative PCR measurements repeated twice. Models are specified in Supplementary Methods and all data and code required to reproduce the analysis are available open-source (see "Data availability" section).

**Reporting summary.** Further information on research design is available in the Nature Research Reporting Summary linked to this article.

## Code availability

All code is available as a static snapshot at the ETH Research Collection with identifier [https://doi.org/10.3929/ethz-b-000471160][41] and in version-controlled form at https://gitlab.com/csb.ethz/topodesign_decoder.

## Data availability

All computational and experimental data that support the findings of this study are available at the ETH Research Collection with the identifier [https://doi.org/10.3929/ethz-b-000471160][41]. Strains and plasmids used in this study are available from Addgene (https://www.addgene.org/browse/article/28211930/). Any other relevant data are available from the authors upon reasonable request.

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

## Acknowledgements
We thank A. Azizoglu for experimental support, L. Widmer and H.-M. Kaltenbach for discussions. This work was supported by the Swiss National Science Foundation via the NCCR Molecular Systems Engineering (grant 182895).

## Author contributions
C.L., F.R., and J.S. conceived the study. C.L. and J.S. conceived TopoDesign. C.L. and F.R. designed experiments. C.L. performed modeling, experiments, and data analysis. C.L. and J.S. wrote the manuscript.

## Competing interests
The authors declare no competing interests.
