## [Peer Review File · Nature Communications]

Reviewers' Comments:

Reviewer #1:

Remarks to the Author:

Lormeau et al present TopoDesign, a fascinating framework for the prediction of novel synthetic gene circuits, using a computational approach that allows them to explore the feasibility of a given circuit, tied to dynamical properties of those circuitries selected to be tested. The work is of increasing value in synthetic biology and is a step in the direction of novel, unbiased, and data informed circuit design. A few questions do arise in the manuscript that I believe need to be addressed.

- It is unclear to me where the parts catalog is derived from. One of the great needs in synthetic biology is the identification of novel parts and, though outside the scope of TopoDesign, it would be important to highlight where the parts come from and how TopoDesign will adapt to identification of novel circuit components/parts (either experimentally or computationally)
- Given the point above, are you not limited to the types of circuits that you can identify? It is unclear how you will generate novel circuits with complex dynamics without an overly complex circuit, if the library of parts is limited.
- You claim that TopoDesign allows for fast prototyping of novel circuits. How fast is this prototyping? I.e., how long does it take TopoDesign to make predictions on a set of possible circuits to generate a predictive set of testable hypotheses?
- The use of small informative networks is smart and can help weed out circuits that are non-functional or do not exhibit the desired dynamics. But when you test your circuits, you end up working within these small informative network space. To address this: can you extend the observation on small networks to highly complex models? and how large can your models get?
- It is not clear how the estimates of the posterior probabilities in your small informative networks will extend to larger models with a potentially larger set of parameters.
- You chose to work with T39 as opposed to T93, given that 39 was more robust and less complex (fewer interactions). Though this is an acceptable process as you design the circuit to test experimentally, it is not evident why a model with fewer interactions should be selected out. Can a more unbiased selection criteria be added to the model selection, incorporating robustness, feasibility, model complexity, and other metrics into a ranked score? This seems highly subjective and seems like cherry picking.
- Of all the topologies possible, your model always picked those that had incoherent feed forward loops. Do you have a reasoning as to why? Is this driven by the data that you have used to optimize your parameters and, consequently, forcing the Bayesian model selection to "move" into a particular circuit space, or is it, as you claim, a design principle? Though this statement can be forward looking, I am not entirely convinced of it, as you don't show experimental data supporting it.
- I didn't understand why you needed to place parameters in 3 different classes (tunable, partially tunable, non-tunable). "Tunability" should be an intrinsic property of the parameter. If the goal is to pre-select circuits with a larger proportion of tunable parameters as opposed to non-tunable ones (or vice versa), then a "tunability score" could be used per circuit, simplifying which circuits to test further.
- It is unclear how the parameter sets are selected. In principle, you could have a parameter set that has 20 tunable parameters and one that has 1 tunable one and 19 non-tunable to achieve the same dynamics. How do you select the optimal parameter sets?

- Your set of differential equations has 4 equations, but it seems that you could simplify FP and FPimm into a single equation, thereby reducing the number of parameters in the model. Why do you explicitly model FPimm?

As a minor remark, the notation on your differential equations is confusing. In particular, I would suggest rewriting the subscripts, as A12TF1 (and similar) is very confusing.

Reviewer #2:

Remarks to the Author:

Summary

Cellular events, such as differentiation and division, are often induced by environmental cues and through modulation of signal dynamics. While many of the pathways have been discovered and characterized, the mechanisms cells use to process transient signals remain elusive. To uncover the design principle of such mechanisms, Lormeau et al. rationally designed gene circuits capable of decoding short pulses while ignoring sustained signals for the yeast mating pathway.

TopoDesign, a computational tool based on approximate Bayesian computation, was critical for the designing process. The authors started with a master network that generates 4,122 possible circuit architectures and used TopoDesign to select for circuits with the desired behaviors. With the robustness and feasibility metrics, they selected for suitable biological parts, constructed and tuned the circuit dynamics. Eventually, they were able to construct a circuit that decodes short pulse signals. Through analyzing the common modules of the viable topologies, the authors discover that incoherent feedforward may be vital in decoding transient cues. They also provided experimental demonstration of the chosen circuit design.

General Comments

1. The authors place a lot of emphasis on the in-silico circuit design process. However, this is largely a technical integration of the strategies that have been well established, including extensive evaluation of network topologies and exploration of optimal parameter choices for each topology. This process is implied in typical gene circuit design and optimization, though it has been more explicitly presented in a few studies. When it comes to parameter choice, both typical sensitivity analysis and bayesian statistics have been used for each model formulation.

Also, TopoDesign is based on a previously published algorithm, named TopoFilter. However, it is not clear from the manuscript what its major improvement/adaptation is, compared to TopoFilter. A lot of the analysis presented here appears conceptually similar to:

Lormeau et al. "Multi-objective design of synthetic biological circuits." IFAC-PapersOnLine 50.1 (2017): 9871-9876.

2. The authors presented extensive results on estimating parameter distributions, which in turn are used to gauge the feasibility and robustness of each circuit topology. However, it is not clear how these parameter distributions map to the experimental system. In particular, for each network architecture, different model formulations are possible. Each formulation can be associated with different parameter combinations. Thus the estimated PDFs for different parameters are also conditional on the model formulation, in addition to the network topology. I think the presentation of the computational analysis runs the risk of over interpreting modeling results.

3. Methodology aside, the ability of IFF (with right parameters) to respond to transient but not sustained signals has been demonstrated previously. For example, see Zhang et al, PLoS CB 2016

(Processing Oscillatory Signals by Incoherent Feedforward Loops), which should be cited. In my view, the major conceptual contribution is the experimental implementation and analysis, which should be significantly enhanced. As it stands, the experimental demonstration of the circuit function is too preliminary.

Other specific comments

1. As noted above, the experimental demonstration of signal processing by their circuit is the major novel contribution. I believe this part should be significantly expanded and improved. Overall, the authors have presented their work in a coherent manner, but the actual performance of the circuit is suboptimal. For example, while their modeling illustrates how circuit response was suppressed by long pulses, their experiment only shows moderate suppression (Figure 4D).

2. This research shows that an incoherent feedforward exists in all viable topologies pointing to a common design principle. To what extent do the authors think this conclusion may help us identify/understand natural transient signal decoder? Is there any rational interpretation of why IFF can contribute to decoding transient signals? Is the complexity of the candidate topologies comparable to the plausible natural pathways? If the complexity increases, do the authors still expect to find such common motif? More analysis may be necessary to strengthen the conclusion and increase the significance of the work.

3. They show that their top choice (T39 = IFF combined with PF) performs better than alternative motifs containing IFF and provided some modeling evidence (Extended Data Figure 3), it is unclear if the better performance also depends on specific model formulation. Also, to better demonstrate the power of their design framework, it's useful to show the better performance experimentally as well.

Reviewer #1 (Remarks to the Author):

Lormeau et al present TopoDesign, a fascinating framework for the prediction of novel synthetic gene circuits, using a computational approach that allows them to explore the feasibility of a given circuit, tied to dynamical properties of those circuitries selected to be tested. The work is of increasing value in synthetic biology and is a step in the direction of novel, unbiased, and data informed circuit design. A few questions do arise in the manuscript that I believe need to be addressed.

We thank you for these positive comments, and for raising the detailed questions below.

R1.1: It is unclear to me where the parts catalog is derived from. One of the great needs in synthetic biology is the identification of novel parts and, though outside the scope of TopoDesign, it would be important to highlight where the parts come from and how TopoDesign will adapt to identification of novel circuit components/parts (either experimentally or computationally).

We agree that expanding to the parts catalog represents a great need in synthetic biology and limits our abilities to construct novel circuits. With this in mind, TopoDesign follows a different logic than other design frameworks such as Cello for logical circuit design, which design circuits by selecting parts from a catalog. We start from abstract parts (known categories such as transcriptional repressors, without specifying quantitative features such as repressor strength) and use Bayesian model selection to derive constraints on these quantitative features (given by the viable parameter spaces of each topology). According to the constraints, we then select suitable parts, or identify that molecular engineering is required to meet the feature requirements.

To clarify this logic, we have modified the text as follows:

- (i) By adding the following statement to subsection 'Topological design framework', 1st paragraph: "At this stage, the biological parts are hypothetical – they exist in principle, but we do not know if specific instances with suitable quantitative characteristics such as repressor strength are available or will need to be established by molecular engineering."
- (ii) By revising the end of the 2nd paragraph to: "Without requiring a catalog of specific biological parts, it explores possible combinations of hypothetical parts. TopoDesign accounts for uncertain knowledge about the parts and their biological variability, which allows capitalizing on existing biological parts that may be ill-characterized."
- (iii) By starting the subsection 'Specification of biological parts and rapid prototyping' as follows: "To specify biological parts, in principle, one can select parts from established catalogs that are expanding in scope also for *S. cerevisiae*^{19,25}. The bottleneck is that quantitative parts characterizations that allow checking if parts fulfill the requirements on parameters for circuit function (via the viable spaces) remain sparse. TopoDesign, however, can also consider less well-characterized components by explicitly accounting for parameter uncertainties. We decided to use our available biological components; we

matched them to model predictions by identifying recurrent constraints on parameters in circuits with ideal feasibility > 0.9 (Fig. 1D, Extended Data Fig. 4).”

R1.2: Given the point above, are you not limited to the types of circuits that you can identify? It is unclear how you will generate novel circuits with complex dynamics without an overly complex circuit, if the library of parts is limited.

As argued above, limitations are rather not in the types of circuits we can identify, but in their possible implementations because parts with the required characteristics may not exist. The model-based analysis of tuning possibilities in the context of the entire circuit, however, alleviates this restriction to a certain extent, compared to other design frameworks (please also see below to tenability). In particular, as our decoder demonstrates, it can allow for circuit function without the comparatively high experimental efforts for the molecular engineering of parts.

R1.3: You claim that TopoDesign allows for fast prototyping of novel circuits. How fast is this prototyping? I.e., how long does it take TopoDesign to make predictions on a set of possible circuits to generate a predictive set of testable hypotheses?

We do not have exact timings for generating the predictions on a set of possible circuits (i.e., for the initial decoder design, Fig. 1), but the compute time was approximately 24 hours on 20 cores.

R1.4: The use of small informative networks is smart and can help weed out circuits that are non-functional or do not exhibit the desired dynamics. But when you test your circuits, you end up working within these small informative network space. To address this: can you extend the observation on small networks to highly complex models? and how large can your models get?

Thank you for the positive comment. We use the small informative networks to estimate parameters of parts, independent of the larger potential decoder circuits they are embedded in, and then propagate this information to all candidate circuits (to update the feasibility in a Bayesian sense). For the evaluation of circuit topologies this allows us to work always in the space of the circuits we could test experimentally. We have now added a corresponding clarification in subsection ‘Specification of biological parts and rapid prototyping’, 2nd paragraph.

Regarding scalability to larger models, the bottleneck will be to sample parameter spaces and filter model topologies based on these samples. For analysis applications, we have previously shown that (i) topological filtering can evaluate large numbers of candidate topologies (e.g., 250’000 in one example; Rybinski et al., BMC Bioinformatics 2020) and (ii) the underlying sampling method can handle 20-30 dimensional parameter spaces (here and Zamora-Sillero et al., BMC Syst Biol. 2011) as well as larger test cases with ~50 dimensions (unpublished). We expect that TopoDesign will be applicable to the design of circuits of similar complexity, but the success of the method may be problem dependent. Scaling is clearly an open issue for further investigation, and we added a corresponding statement to the Discussion.

R1.5: It is not clear how the estimates of the posterior probabilities in your small informative networks will extend to larger models with a potentially larger set of parameters.

Please see our answer to R1.4 on how we transfer the posterior probabilities to any network in principle. In at least two cases, however, this extension is not straightforward in practice:

- (i) it is not possible or experimentally efficient to construct and test a set of small networks that cover all parts (parameters); in this case, one can use prior parts characterizations if available (and corresponding uncertainties) or uniform distributions for the missing parameters in the Bayesian update; and
- (ii) small, manually designed networks used for parts characterization are not sufficiently informative; future extensions involving model-based experimental design approaches may address this issue.

For (ii), we have included this perspective in the Discussion.

R1.6: You chose to work with T39 as opposed to T93, given that 39 was more robust and less complex (fewer interactions). Though this is an acceptable process as you design the circuit to test experimentally, it is not evident why a model with fewer interactions should be selected out. Can a more unbiased selection criteria be added to the model selection, incorporating robustness, feasibility, model complexity, and other metrics into a ranked score? This seems highly subjective and seems like cherry picking.

We thank you for pointing out that we did not provide the complete rationale for selecting T39 over T93. The revised passage that we hope clarifies this point now reads: “Two circuits, T39 and T93, stood out by having close to 50% feasibility. T39 (Fig. 4C) was more robust and had fewer interactions than T93 (Extended Data Fig. 2). Importantly, the additional induced degradation of LexA in T93 is likely to increase the circuit’s burden on the cell and thereby to make it evolutionarily less stable²⁷. Finally, tuning without molecular engineering to modify parameters such as the cooperativity of LexA should make T39 a functional decoder (Fig. 4D, Extended Data Figs. 6-7). We therefore selected T39 for implementation.”

More generally, we considered a combined metric for model selection during the development of TopoDesign but decided against it primarily because it would hide the tradeoffs between robustness, feasibility, complexity, and potentially other criteria in an arbitrary manner. In practice, a ranked score would amount to a linear combination of partial scores akin to scalarization in multi-objective optimization, presenting one point of a Pareto front. In addition, the ranking would rely on the weights for the combination, and we do not know of an unbiased way of deriving them. We therefore suggest that the user should evaluate the tradeoffs, such that subjective choices are explicit.

R1.7: Of all the topologies possible, your model always picked those that had incoherent feed forward loops. Do you have a reasoning as to why? Is this driven by the data that you have used to

optimize your parameters and, consequently, forcing the Bayesian model selection to "move" into a particular circuit space, or is it, as you claim, a design principle? Though this statement can be forward looking, I am not entirely convinced of it, as you don't show experimental data supporting it.

Our first line of evidence to believe that incoherent feed forward loops are indeed required for decoder function (i.e., a possible design principle, based on the space of possible circuits we investigated) relies on the search for functional topologies being unbiased by experimental data. Within fixed parameter bounds (admitting values over several orders of magnitude), our Bayesian model selection operated with a uniform prior and should therefore in theory be unbiased. We incorporated experimental data from the small informative circuits at a later stage. The revised text contains a corresponding statement for clarification (1st paragraph of subsection 'Topologies for a short pulse decoder').

Regarding our second line of evidence, and to explain the rationale for incoherent feed forward loops in the short pulse decoder designs, we have added Figure 2 to the main text (formerly Fig. S3), a paragraph in subsection 'Topologies for a short pulse decoder', and a new analysis using simplified models based on network motifs (Supplementary Methods, section 'Network motifs and their combination to functional decoders' and Extended Data Fig. 3). The new analysis confirms the findings from TopoDesign. It also demonstrates a design not found in our topology search (because of biological implementation constraints in gene networks) relying on an overall structure of an incoherent feed forward with embedded time-delayed negative feedback (please see also our response to R2.5 for details)

R1.8: I didn't understand why you needed to place parameters in 3 different classes (tunable, partially tunable, non-tunable). "Tunability" should be an intrinsic property of the parameter. If the goal is to pre-select circuits with a larger proportion of tunable parameters as opposed to non-tunable ones (or vice versa), then a "tunability score" could be use per circuit, simplifying which circuits to test further.

Indeed, 'tunability' is an intrinsic and quantitative property of a parameter, essentially reflecting the experimental effort associated with changing a parameter's value (in a given range). We agree that this property is hard to quantify in general (e.g., depending on parts availability if, for example, promoters are concerned where libraries of parts with different expression strength may exist, and depending on a particular lab's experimental capabilities).

Our classification considers a combination of this 'intrinsic tunability' and of how TopoDesign handles the different parameter classes:

- (i) Fully tunable parameters: Experimentally, we have total control over these parameters, without molecular engineering (e.g., the aTc concentration). In TopoDesign, fully tunable parameters have no uncertainty. We can also vary them experimentally (to measure dose responses) during the small network characterization to obtain more information on the

other parts. To compute the updated feasibility, which represents a best-case scenario after parameter tuning, we optimize fully tunable parameters.

- (ii) Partially tunable parameters: Those parameters can be tuned quite easily by molecular engineering (e.g., copy numbers). Partially tunable parameters are associated with uncertainty; hence, they need to be in the set of estimated parameters. In addition, their values are optimized to compute the updated feasibility.
- (iii) Non-tunable parameters: These are assumed to be fixed because of high experimental efforts for changing their values (e.g., affinities), their values need to be estimated, and they cannot be varied during the optimization to compute the updated feasibility.

Because we selected circuits according to feasibility after (predicted) optimal tuning of each circuit, and because our feasibility metric already includes a potential positive effect of tunability, we did not include a separate tunability score.

The revised text (section 'Selection of biological parts and rapid prototyping', 2nd paragraph) now explains this rationale as follows: "We used the ABC posterior to update the feasibility of all circuits (Fig. 4A; Extended Data Fig. 2). Because the non-zero regions of the posteriors did not overlap with the topologies' viable spaces, all circuits had zero updated feasibility. To increase feasibility, we allowed for tuning of selected parameters. Tunability is an intrinsic property of a parameter, reflecting the experimental effort for modifying a parameter's value in given ranges. We devised discrete categories of tunability that account for this effort and have different roles in TopoDesign (Supplementary Methods). As simple tuning possibilities to increase feasibility, we considered varying aTc concentrations and promoter copy numbers (assuming those affect maximum production rates proportionally)."

In addition, we included the extended explanation for the classification of parameters above in subsection 'Topological Design Framework' of the Supplementary Methods.

R1.9: It is unclear how the parameter sets are selected. In principle, you could have a parameter set that has 20 tunable parameters and one that has 1 tunable one and 19 non-tunable to achieve the same dynamics. How do you select the optimal parameter sets?

Once the parameters are assigned to tunable, partially tunable, or non tunable, categories do not change over the whole design process. So two parameter sets of the same circuit always have the same number of tunable and non-tunable parameters. For a given circuit, we optimize the parameters in the tunable dimensions (the same for all parameter sets) to get the desired dynamics. Then for each circuit, we check the probability of a parameter set to achieve the right dynamics after tuning, and we call it the feasibility of the circuits. Now indeed, it is possible in principle that two circuits reach the same feasibility, one after tuning 20 parameters and the other after tuning only one parameter. If this is the case, and if this is the highest feasibility of all circuits, the user would naturally select the circuit that requires less tuning effort.

R1.10: Your set of differential equations has 4 equations, but it seems that you could simplify FP and FPimm into a single equation, thereby reducing the number of parameters in the model. Why do you explicitly model FPimm?

We modeled the immature FP explicitly because the time delay between protein production and observation of fluorescence of ~60 mins for citrine (Supplementary Table 5) is significant for our circuits that operate on similar time-scales (e.g., regarding the pulse durations). The corresponding model parameter is fixed to literature values; for the purpose of design including FP maturation explicitly does not affect model complexity.

R1.11: As a minor remark, the notation on your differential equations is confusing. In particular, I would suggest rewriting the subscripts, as A12TF1 (and similar) is very confusing.

We have now changed the notation for the subscripts to make it more intuitive while keeping it precise (complicated subscripts result from the need for a different $K_{m_{P2x-TFy}}$ for each possible interaction, instead of a single K_m in the Hill functions). We have added a corresponding short explanation to the Supplementary Methods.

Reviewer #2 (Remarks to the Author):

Summary

Cellular events, such as differentiation and division, are often induced by environmental cues and through modulation of signal dynamics. While many of the pathways have been discovered and characterized, the mechanisms cells use to process transient signals remain elusive. To uncover the design principle of such mechanisms, Lormeau et al. rationally designed gene circuits capable of decoding short pulses while ignoring sustained signals for the yeast mating pathway.

TopoDesign, a computational tool based on approximate Bayesian computation, was critical for the designing process. The authors started with a master network that generates 4,122 possible circuit architectures and used TopoDesign to select for circuits with the desired behaviors. With the robustness and feasibility metrics, they selected for suitable biological parts, constructed and tuned the circuit dynamics. Eventually, they were able to construct a circuit that decodes short pulse signals. Through analyzing the common modules of the viable topologies, the authors discover that incoherent feedforward may be vital in decoding transient cues. They also provided experimental demonstration of the chosen circuit design.

General Comments

R2.1: The authors place a lot of emphasis on the in-silico circuit design process. However, this is largely a technical integration of the strategies that have been well established, including extensive evaluation of network topologies and exploration of optimal parameter choices for each topology. This process is implied in typical gene circuit design and optimization, though it has been more explicitly presented in a few studies. When it comes to parameter choice, both typical sensitivity analysis and bayesian statistics have been used for each model formulation.

Also, TopoDesign is based on a previously published algorithm, named TopoFilter. However, it is not clear from the manuscript what its major improvement/adaptation is, compared to TopoFilter. A lot of the analysis presented here appears conceptually similar to: Lormeau et al. "Multi-objective design of synthetic biological circuits." IFAC-PapersOnLine 50.1 (2017): 9871-9876.

Regarding the emphasis on the in-silico design process in terms of extent of text, we feel that substantial shortening would rather obscure key concepts of the methodology we propose (please see also our answers to Reviewer 1, which led to further clarifications in the revised manuscript).

We agree with the reviewer that strategies such as 'extensive evaluation of network topologies and exploration of optimal parameter choices for each topology' and '[for parameter choices] sensitivity analysis and bayesian statistics [...] for each model formulation' are well-established. Such strategies relying on enumeration of topologies, however, do not scale well when the number of possible topologies is high. TopoDesign achieves scaling by simultaneously exploring

topologies and parameters (essentially by parametrizing topologies and re-using parameter samples for different, nested topologies), which is conceptually different. To clarify this distinction we have revised the related statement in subsection ‘Topological design framework’, 2nd paragraph to: “Also, existing computational design methods either help designing dynamic circuits for few topologies that can be enumerated and analyzed individually¹⁸, or focus on logical circuits using well-characterized components^{17,19}.”

Indeed, we developed TopoDesign based on TopoFilter (for Bayesian model selection in systems biology analyses), with the basic ideas for the framework presented in the IFAC paper the reviewer cites. We hope that the reviewer agrees with the description of this evolution in the manuscript as: “... we extended our Bayesian circuit design method²⁰ to a rapid prototyping method, TopoDesign.”

At a high level, the IFAC paper presents parts of the TopoDesign framework (panels A,B in Extended Figure 1) in a purely in-silico setting that ends with ranking circuit topologies by complexity, robustness, and feasibility. To achieve integration of experimental data, rapid prototyping, and eventually construction of a functional decoder, several extensions and modifications of the design method presented in the IFAC paper were necessary. This includes:

- (i) Modification of the feasibility metric to account for the multidimensional shapes of viable spaces: previously, we used min/max bounds on the marginals.
- (ii) Formulation of the feasibility metric in terms of probability distributions over parameter spaces: This conceptual difference is critical because it enables a fully Bayesian framework with determination of ideal feasibility (best-case scenario, with realistic parameter uncertainties) as well as real feasibility (after incorporating posteriors from characterization experiments).
- (iii) Introduction of tuning possibilities: Again via the feasibility metric (in an optimization setting) this feature allows to achieve functionality of the (decoder) implementation with reduced experimental effort.
- (iv) As a consequence of the above, and with additional extensions (such as statistics over topologies to suggest suitable initial parts; construction of small informative circuits), development of the full TopoDesign framework shown in Extended Figure 1.

To make this distinction from our previous work clearer, the revised text (subsection Topological design framework’, 3rd paragraph) now reads: “To account for such dependencies between parts in practice, we therefore measure feasibility: the proportion of a parameter distribution that fits into the viable space (Fig. 1C). This formulation of the metric is distinct from²⁰ and critical: it enables a systematic integration of experimental data, and thereby all iterations of computation and experiments.”

R2.2: The authors presented extensive results on estimating parameter distributions, which in turn are used to gauge the feasibility and robustness of each circuit topology. However, it is not clear how these parameter distributions map to the experimental system. In particular, for each

network architecture, different model formulations are possible. Each formulation can be associated with different parameter combinations. Thus the estimated PDFs for different parameters are also conditional on the model formulation, in addition to the network topology. I think the presentation of the computational analysis runs the risk of over interpreting modeling results.

The reviewer is entirely correct that all inferences and predictions depend on the model formulation, that is, the specific functional forms in our ODE model that capture the processes and interactions. Note that, across topologies, the corresponding terms and therefore parameters and their meaning are consistent; we have strictly nested models.

Unfortunately, we do not know of a universal and systematic way of defining these terms without extensive prior knowledge and data. We therefore used formulations that are standard in the field because they proved suitable across many different applications, have interpretable parameters that relate to experimental observations, and are sufficiently flexible for our design purposes (e.g., Hill functions for gene expression control at promoters). A risk of over-interpreting modeling results remains, but we argue that it is low in our context because we iteratively validate model predictions with experimental analyses, ultimately yielding the functional decoder.

To make the reader aware of this risk, we added the following statement to subsection 'Topological design framework', 3rd paragraph: "For the model, we used commonly applied specifications of processes and interactions, such as Hill functions for gene expression control; note that all model inferences and predictions are contingent on this formulation."

R2.3: Methodology aside, the ability of IFF (with right parameters) to respond to transient but not sustained signals has been demonstrated previously. For example, see Zhang et al, PLoS CB 2016 (Processing Oscillatory Signals by Incoherent Feedforward Loops), which should be cited. In my view, the major conceptual contribution is the experimental implementation and analysis, which should be significantly enhanced. As it stands, the experimental demonstration of the circuit function is too preliminary.

Indeed, it is well-known that IFFs can differentiate between transient and sustained signals, which is why we included this possibility in the master network. Our decoder, however, has a critical design requirement that distinguishes it from previous work we are aware of: the decoder should not respond *at any time point* (and not only in the limit) to an input above a critical duration. This implies, for example, that a response to a sufficiently short pulse can only occur after the duration of the pulse (otherwise, no distinction between short and long pulses is possible) and that memory of the pulse is needed (otherwise a distinction between no pulse and a short pulse is impossible).

In addition to an extended explanation of decoder functions (please see our responses to R1.5 and R2.5), we have added corresponding clarifications and references (including Zhang et al.) in subsection 'Topological design framework', 1st paragraph: "We included IFFs in particular

because they can retain memories of pulses¹⁵, and thereby discriminate between transient and sustained inputs¹⁶.” Note, however, that we require a decoder behavior that is different from the known behaviors of IFFs: it should not respond to a long pulse at any point in time, not only after an adaptation period.”

Regarding the experimental implementation and analysis, we assume that the reviewer refers to the specific comments below – please see our responses there.

Other specific comments

R2.4: As noted above, the experimental demonstration of signal processing by their circuit is the major novel contribution. I believe this part should be significantly expanded and improved. Overall, the authors have presented their work in a coherent manner, but the actual performance of the circuit is suboptimal. For example, while their modeling illustrates how circuit response was suppressed by long pulses, their experiment only shows moderate suppression (Figure 4D).

Regarding the circuit’s performance, we have now expanded the analysis, as detailed in our response to R2.6 below. Briefly, whereas for simpler (e.g., logical) circuits, performance in terms of output signal separation by fold-changes of several orders of magnitude are common, this is not the case for our decoder when cell-to-cell variability in a population is taken into account. All modeling results in the previous version of the manuscript pertained to single ‘best’ cells, and did not consider this variability.

R2.5: This research shows that an incoherent feedforward exists in all viable topologies pointing to a common design principle. To what extent do the authors think this conclusion may help us identify/understand natural transient signal decoder? Is there any rational interpretation of why IFF can contribute to decoding transient signals? Is the complexity of the candidate topologies comparable to the plausible natural pathways? If the complexity increases, do the authors still expect to find such common motif? More analysis may be necessary to strengthen the conclusion and increase the significance of the work.

Regarding a rational explanation of why nested IFFs can contribute to decoding transient signals, we have added Figure 2 to the main text (formerly Fig. S3) and a paragraph in subsection ‘Topologies for a short pulse decoder’. Briefly, the decoding of transient signals requires two types of dynamic behavior: (i) a response to the input, a memory of the input, and an adaptation to the basal level irrespective of input duration; and (ii) a response to the input that prevents an output if the input duration is too short or too long. The interaction of both leads to increasing output after the short input pulse only. In all but one of the decoder designs found by TopoDesign, this behavior is achieved by interlocked IFFs for (i) and coherent feedforwards (CFFs) for (ii).

We thank the reviewer for suggesting more analysis to strengthen this conclusion. We have now included an analysis using simplified models based on network motifs similar in spirit to the

original study of feedforward motif functions¹. It confirms the rationale above. It also demonstrates a design not found in our topology search (because of biological implementation constraints in gene networks). Starting from the fact that IFFs as well as three-node, time-delayed negative feedbacks (NFs) can both achieve biochemical adaptation², we devised an alternative architecture that relies on an overall structure of an IFF with an embedded time-delayed NF (see Supplementary Methods, new section 'Network motifs and their combination to functional decoders' and Extended Data Fig. 3). Given the general nature of network motifs and our two lines of evidence, we expect these principles to hold also for more complex circuits (as our functional topologies show), but of course this does not preclude that even more complex alternatives without IFFs exist that establish a short pulse decoder function. We state this potential limitation in the revised Discussion (first paragraph).

Regarding natural pathways, we think that our work can help identify and understand transient signal decoders primarily by the generalizations possible in view of network motifs. We have now added specific examples for this in a new part of the Discussion (second paragraph). Briefly, for multicellular eukaryotes it has been demonstrated both that interlocked feedforwards exist downstream of signaling pathways and that short pulse decoding can have a physiological function. In the examples provided, we focus on Erk signaling consistent with our introduction to the topic of decoding transient signals.

R2.6: They show that their top choice (T39 = IFF combined with PF) performs better than alternative motifs containing IFF and provided some modeling evidence (Extended Data Figure 3), it is unclear if the better performance also depends on specific model formulation. Also, to better demonstrate the power of their design framework, it's useful to show the better performance experimentally as well.

We thank the reviewer for pointing out the important links between power of the design framework and experimental performance of predicted decoder circuits.

Unfortunately, we could not follow the detailed comment entirely because we considered potential measures of performance, such as output fold-changes for different inputs, only as necessary criteria for function (via the cost function), and not to rank circuit designs. Correspondingly, we did not make any claims on the relative performance of circuit topologies; the primary aim of the study was to find a functional circuit rapidly. Nonetheless, we included controls demonstrating experimentally that predicted non-functional circuits (without IFF) do not perform as decoders.

Briefly, to address the link between quantitative performance and power of the design framework, we have extended the analysis of T39. Specifically, we evaluated the average predicted performance (in terms of fold-changes of responses of a cell population) for circuit T39 with optimal copy numbers and aTC concentrations. This represents a best-case scenario for circuit performance. The comparison of experimental and simulated performance revealed that the former is within a margin of the best case that we argue is reasonable for a complicated dynamic

system; especially with gene expression noise, which our models do not capture, we would expect a diminished performance. We have revised the complete subsection 'Circuit implementation and validation' accordingly, including figures 5 and 6 with additional data and simulation results. For details, we refer to this part of the manuscript.

References

- 1 Mangan, S. & Alon, U. Structure and function of the feed-forward loop network motif. *Proceedings of the National Academy of Sciences* **100**, 11980-11985, doi:10.1073/pnas.2133841100 (2003).
- 2 Ma, W., Trusina, A., El-Samad, H., Lim, W. A. & Tang, C. Defining Network Topologies that Can Achieve Biochemical Adaptation. *Cell* **138**, 760-773, doi:10.1016/j.cell.2009.06.013 (2009).

Reviewers' Comments:

Reviewer #1:

Remarks to the Author:

The issue of novel circuit design, broadly defined, tied to identification of novel parts and key circuit components, is of paramount interest in synthetic biology. The research presented here is a step in that direction, where computational models will allow us to explore the complex landscape of circuit feasibility.

The authors have addressed all the comments/concerns that I presented satisfactorily and I recommend the work for publication.

Reviewer #2:

Remarks to the Author:

The authors have satisfactorily addressed/clarified the several issues I raised in my original comments:

- 1) Connection with and advances over previous work, including their own.
- 2) Clarification on the model formulation and limitation of the computational framework in identifying potential circuit candidates.
- 3) Further analysis and clarifications on the experimental demonstration.

As a result, I believe the work has been significantly improved.